# Dynamic Stance:
# Modeling Discussions by Labeling the Interactions

◇***Blanca Calvo Figueras** ◇**Irene Baucells** ⊙**Tommaso Caselli**
◇Barcelona Supercomputing Center
*HiTZ Center - Ixa, University of the Basque Country UPV/EHU
⊙CLCG, University of Groningen, The Netherlands
◇*`blanca.calvo@ehu.eus`, ◇`irene.baucells@bsc.es`
⊙`t.caselli@rug.nl`

## Abstract

Stance detection is an increasingly popular task that has been mainly modeled as a static task, by assigning the expressed attitude of a text toward a given topic. Such a framing presents limitations, with trained systems showing poor generalization capabilities and being strongly topic-dependent. In this work, we propose modeling stance as a dynamic task, by focusing on the interactions between a message and their replies. For this purpose, we present a new annotation scheme that enables the categorization of all kinds of textual interactions. As a result, we have created a new corpus, the Dynamic Stance Corpus (`DySC`), consisting of three datasets in two middle-resourced languages: Catalan and Dutch. Our data analysis further supports our modeling decisions, empirically showing differences between the annotation of stance in static and dynamic contexts. We fine-tuned a series of monolingual and multilingual models on `DySC`, showing portability across topics and languages.

## 1 Introduction

Stance detection consists of identifying opinions, perspectives, or attitudes expressed in texts. The task has been found to be relevant as a component for fact verification, argument mining, and media analysis (Boltužić and Šnajder, 2014; Baly et al., 2018; Bauwelinck and Lefever, 2020; Guo et al., 2022), but also on its own (Hardalov et al., 2022). The standard setting for stance detection is the adoption of a static perspective, where different texts are labeled with a three-class label approach, i.e., being in *favour*, *neutral* or *against* a given topic (Somasundaran and Wiebe, 2009; Mohammad et al., 2016; Conforti et al., 2020; Clark et al., 2021, amog others). One of the limits of this approach is its strong dependence on topics, resulting in poorly portable systems (Reuver et al., 2021).

Other works have taken a different direction: rather than keeping a fixed topic, stance is investigated between pairs of texts. Key contributions in this direction are the RumourEval shared tasks (Derczynski et al., 2017; Gorrell et al., 2019), the Fake News Challenge (Pomerleau and Rao, 2017; Hanselowski et al., 2018) and, more recently, the Stanceosaurus corpus (Zheng et al., 2022a). Although different with respect to the nature of the data, in all cases systems are required to determine the stance of a message with respect to a previous claim with the goal of annotating stance as a component of fact or rumor verification. Consequently, the used labels focus on denying or supporting some facts instead of modeling the actual internal dynamics of an online exchange.

**Our contributions** In this work, we present a new annotation scheme to model stance dynamically as a way to account for the evolution of an online exchange between two users. We introduce Dynamic Stance detection as a new independent task, which captures different insights from Static Stance detection. To this end, we have manually annotated a new multilingual corpus with short exchanges between pairs of users from different social media platforms. We have worked with two middle-resourced languages, namely Catalan and Dutch, and conducted an extensive set of experiments investigating the portability of the trained models across topics and languages. We also show the relation between Dynamic Stance and Static Stance labels.

The remainder of the paper is structured as follows: in Section 2, we present a critical overview of previous work to highlight the differences and innovations of our proposal. Section 3 introduces the Dynamic Stance task and presents our annotation scheme. The Dynamic Stance Corpus (`DySC`) is discussed in Section 4, focusing on the data collection, the annotation process, and analyzing the relation between Static and Dynamic Stance. Our experiments and results are presented in Section 5.

Finally, Section 6 summarises our findings and discusses future work.

## 2 Related Work

Stance has been modeled extensively as a static task in which a text is tagged with respect to its attitude towards a specific topic or target (Mohammad et al., 2016; Küçük and Can, 2020; Cao et al., 2022). Methods for stance classification have evolved alongside the developments in Natural Language Processing, moving away from feature-based approaches up to fine-tuning and prompting pre-trained models (Ferreira and Vlachos, 2016; Aker et al., 2017; Fang et al., 2019; Zhang et al., 2020; Allaway and McKeown, 2020; Zheng et al., 2022b, among others). While performances on benchmark data have seen improvements, issues such as balance among classes and portability of systems across topics and languages remain open research questions.

Stance detection has only been approached as a relation between two texts on a few occasions. As already mentioned, remarkable contributions are the Fake News Challenge (Pomerleau and Rao, 2017), RumourEval (Derczynski et al., 2017; Gorrell et al., 2019), and the Stanceosaurus corpus (Zheng et al., 2022a).

In the Fake News Challenge, the authors aim at using disagreements between news pieces to detect false information. They design an annotation scheme based on Ferreira and Vlachos (2016) to estimate the stance of a body text from a news article relative to a headline in English. The labels they define are *agrees, disagrees, discusses*, and *unrelated*. While the headline and the content of an article are very often in agreement, in online debates this is not frequent. Therefore, more granular labels are needed to model messages engaging with each other in an exchange of opinions.

In RumourEval, the goal is to verify a rumor in English by modeling the discourse around it. From this perspective, the authors designed an annotation scheme that focuses on making explicit whether the replies to a rumor on Twitter support or deny the given claim. Four labels are available: *support, deny, comment*, and *query*. The Stanceosaurus corpus (Zheng et al., 2022a) follows a similar approach. In this work, the authors provide specific claims and look for Twitter messages related to those claims. The corpus is multilingual, covers multiple topics, and uses a more fine-grained set of labels when compared to RumourEval (*supporting, refuting, discussing, querying*, and *irrelevant*). Stanceosaurus is the largest available corpus for stance with respect to claims. Besides the size of the corpus, the authors obtain a macro F1 score of 0.53 when testing on unseen claims,[1] with performances dropping to 0.40 for zero-shot cross-lingual transfer in Hindi and Arabic.[2]

The focus of these two studies was to use interactions between texts to verify claims. Therefore, they use labels such as *support* and *deny*, which entail that the departing message must be a claim, which is the case in RumourEval and Stancesaurus. However, in online interactions, most of the messages we encounter are not claims but rather opinions and perspectives, which can not always be supported or denied, but rather agreed upon or disagreed.

When it comes to language-specific resources for stance detection in Catalan and Dutch, previous work is quite limited and focused on modeling Static Stance. For Catalan, stance is investigated in Twitter and only with respect to the "Catalan Indepedence" topic (Taulé et al., 2017, 2018; Zotova et al., 2020). For Dutch, the only available dataset is Wang et al. (2020), where stance is annotated targeting face masks and social distance measures. The corpus comprises messages from different platforms (Twitter, Reddit, and Nu.nl). Additionally, the CoNTACT corpus (Lemmens et al., 2022) identifies stance towards vaccines using Twitter and Facebook posts, but their data is not publicly available.

Our work departs from previous studies by focusing on modeling online exchanges intrinsically. The goal of this approach is to be able to capture the relation of a text with respect to a disputed claim as well as any other dialogic interaction in online forums or micro-blogging platforms. This will be relevant for tasks such as fact verification, but also for controversy detection (Hessel and Lee, 2019; Figueras et al., 2023), argument mining (Lawrence and Reed, 2020; Dutta et al., 2022; Ruiz-Dolz et al., 2022), detecting previously verified claims (Nakov et al., 2018; Shaar et al., 2022; Nakov et al., 2021), and discourse analysis.

---

[1]Fine-tuning a BERT-large model (Devlin et al., 2019)
[2]Fine-tuning a mBERT model (Devlin et al., 2019)

## 3 Dynamic Stance Annotation

Inspired by this previous work, we hypothesize that the stance of a message towards a previous message (Dynamic Stance) can show more linguistic patterns than the static approach, as it will be less topic-dependent than modeling agreement over a topic. While Dynamic Stance is an independent task, for the purpose of this work, we also annotated Static Stance. Having access to both annotation levels is essential to empirically investigate the interactions between these two schemes, showing similarities and differences.

**Static Stance**    As highlighted in Section 2, previous work has mainly targeted stance annotation in isolation, i.e., with respect to a given topic. Annotation efforts on Static Stance are very similar to each other, mainly differentiating for the topics and text types (Conforti et al., 2020). To maximize compatibility and reusability of data from previous work, we inherit the labels and definitions proposed by Mohammad et al. (2016). In particular, we have a four-label annotation scheme:

- **Favour**: The message expresses a positive stance or opinion with respect to the topic;

- **Against**: The message expresses a negative stance or opinion with respect to the topic;

- **Neutral**: The message is about the topic but does not hold any recognizable stance, e.g., it can report facts about the given topic;

- **NA** (Not Applicable): The message may or may not express a stance, but its content does not address the target topic; or the message is intelligible.

**Dynamic Stance**    To capture the dynamic dimension of stance, the target message must always be put in relation to its direct parent, that means, the message that the target message is replying to. Dynamic Stance annotation is inherently a two-step process: first, the annotators have to read the parent and the reply messages, secondly, they must assign them a label by answering the following question: "*What is the stance or opinion of the reply with respect to its parent?*". To properly answer this question, the set of labels must be richer but, at the same time, manageable, i.e., easy to understand and to remember. To define this set we have departed from the labels in Pomerleau and Rao (2017), and extended them to the granularity of Zheng et al. (2022a). We have identified seven labels:

- **Agree**: The reply agrees with the parent message.

- **Disagree**: The reply disagrees with the parent message. This can include questioning or making fun of the parent message with a clear dislike towards it. If the reply questions the parent message and it shows a clear disagreement with it (e.g. use of irony, rhetorical questions, etc.), the reply will be labeled as Disagree.

- **Elaborates**: The reply is not against the parent message, but it expresses additional opinions/information with which we do not know if the parent message would agree.

- **Query**: The reply will be annotated as Query if the message questions or requires more arguments/proof/evidence about the parent message. It will not be annotated as Query if the question is rhetorical, ironic, or has a clear stance towards the parent message.

- **Neutral**: The reply is on the same topic as the parent message but does not express any position or opinion about it. If the parent message was a genuine question with no opinion (also not ironic or rhetorical), the reply should be labeled as Neutral.

- **Unrelated**: The reply is not on the same topic and it has no relation to the parent message.

- **NA**: One or both of the messages in the pair are unintelligible, e.g., a URL.

In our guidelines, we emphasize that, in the cases in which the parent message does not hold an opinion, the label of the reply must be assigned on the basis of the content. Therefore, if the parent message is a descriptive text with no opinion of its own (e.g. "EMA approves another COVID vaccine"), the stance of the reply toward the content of the text should still be considered, and a reply such as "More poison for us." should be labeled as *Disagree* (i.e. the reply disagrees with the facts described in the parent message but not necessarily with its author, who did not express any opinion).[3]

Figure 1 illustrates an example of these schemes. From a Static Stance perspective, the parent message is against vaccines, while the reply is in favor. At the Dynamic layer, the parent and reply stand

---

[3]The full guidelines can be found here: https://github.com/projecte-aina/dynamic-stance-analysis/tree/main/create_dataset/annotation/guidelines

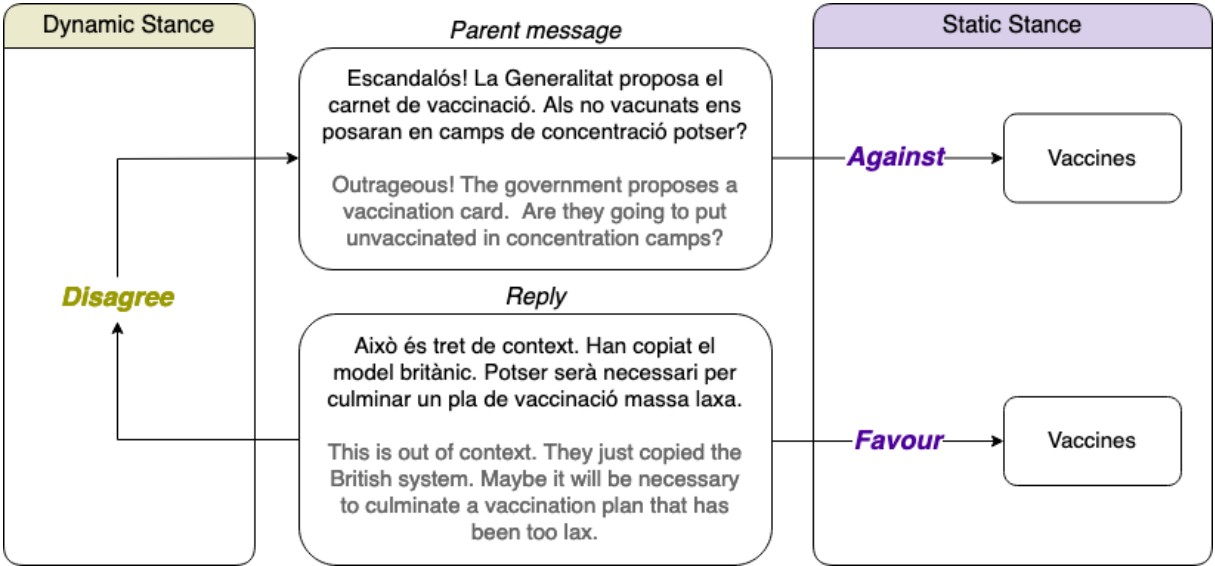

Figure 1: Example of the two annotation schemes. The Dynamic Stance scheme (left) assesses the relation between the reply and the parent message, while the Static Stance scheme (right) identifies the relation of each message with the topic, in this case, vaccines. Messages are extracted from the DySC corpus.

in a disagreement relationship. As we will show in Section 4.2, the dynamic label cannot always be inferred from the static ones.

## 4 The Dynamic Stance Corpus

The Dynamic Stance Corpus (DySC) is a multi-lingual, and multi-topic corpus in Catalan and Dutch. This section will illustrate the methods that were used for collecting and annotating the data in the two languages. We will also analyze the resulting labels.

### 4.1 Data Collection

**Catalan Data Collection** For Catalan, we collected data from two online platforms: Twitter and the Racó Català forum.[4] Given the different nature of the platforms, we have kept the data separated.

The Catalan Stance and Emotions Twitter dataset (CaSET)[5] has been obtained by extracting messages belonging to five different threads. Each thread identifies a controversial topic in Catalan society and it has been identified by means of keywords and limited to specific time periods (see Appendix A for the full list of keywords and time periods per thread). We targeted the following discussion threads: (i) the COVID-19 vaccination campaign; (ii) the regulation of rent prices; (iii) the expansion of Barcelona's El Prat airport; (iv) the legalization of surrogate pregnancy; and (v) the rigging in a TV music contest that decided the Spanish candidate for Eurovision.[6] We used the Twitter Academic API to collect the messages. To maximize the number of messages that contain a reply, we have developed a pipeline that first finds whether a message contains any of the target keywords for a thread during the relevant time period, and subsequently checks if the target message has a parent message. If so, both the parent and the target messages are retrieved and stored. In this way, we collected 11,078 unique messages organized in 6,673 pairs, which have been manually annotated.

For the Catalan Stance and Emotion Racó dataset (CaSERa),[7] we followed a slightly different approach. We collected messages belonging to random online exchanges compliant with the following requirements: (a) exchanges must have a minimal length of two messages and a maximum length of four messages; (b) up to two branches of the same conversation can be collected; and (c) all messages in the exchange have to be between 300 and

---

[4]https://www.racocatala.cat/

[5]This dataset has also been annotated with emotion labels, but this level of annotation is not reported in this paper. Find the data in https://huggingface.co/datasets/projecte-aina/CaSET-catalan-stance-emotions-twitter

[6]https://elpais.com/television/2022-01-30/el-publico-contra-el-jurado-las-reacciones-ante-la-victoria-de-chanel-en-el-benidorm-fest.html

[7]This dataset has also been annotated with emotion labels, but this level of annotation is not reported in this paper. Find the data in https://huggingface.co/datasets/projecte-aina/CaSERa-catalan-stance-emotions-raco

15 characters. The requirements prevent the collection of very long forum posts, which may have contained many opinions, and very long threads, which may have over-represented some topics in the dataset. We obtained 14,000 pairs, which have been manually annotated.

**Dutch Data Collection** We have collected a comparable Twitter dataset for Dutch. The Dutch Stance Twitter dataset (DuST) aims at testing the feasibility of cross-lingual transfer of Dynamic Stance. For this reason, we limited the collection of the data to a common online platform, i.e., Twitter, and a common topic, namely the COVID-19 vaccination campaign. Focusing on the same online platform and a common topic is a strategy to minimize language variety factors that may negatively affect the cross-lingual transfer learning approach (Ramponi and Plank, 2020). To maximize the similarities with CaSET, we have used the same pipeline approach. Wherever possible, we have used translations of the Catalan keywords for the COVID-19 vaccination. However, we integrated the keyword sets with others that mirrored culturally specific aspects of the vaccination in the Netherlands and Belgium (e.g., *3G*, or *coronabewijs*). We obtained 14,840 messages of which 2,485 pairs have been manually annotated.

## 4.2 Data Annotation and Analysis

The data annotation has been conducted by three separate teams. For Catalan, expert annotators have been recruited and trained.[8]

For CaSET, a total of six annotators plus two judges were involved. The judges had the role of resolving disagreements when unanimous annotations could not be reached. The annotation of the Static and Dynamic Stance has been conducted in parallel. A team of two annotators and one judge worked on the Static Stance, while we had a team of four annotators plus one judge for Dynamic Stance. The annotation of Dynamic Stance is more challenging than the static one: for Dynamic Stance, the average Cohen's *kappa* between independent annotators was 0.57, while this score jumps to 0.83 for the static annotation. This means that the judges were involved in 1,284 cases for the Dynamic Stance annotation (18.95% of the instances), and 2,087 for the Static Stance annotation (18.84% of the instances).

---

[8]All annotators had a regular contract and received a fair payment in line with Spanish labor laws.

The CaSERa dataset has been annotated for Dynamic Stance only, lacking any specific topic for the Static Stance. We have followed the same annotation procedure as in CaSET and employed the same annotators. Similarly to the scores obtained in CaSET, the average Cohen's *kappa* is 0.58. The judge was involved in 18.17% of the cases.

For DuST, we have followed a slightly different annotation procedure. A team of three students performed the annotations as part of their thesis project. Similarly to the expert annotators, students received training and went through calibration sessions. As in CaSERa, we have annotated the messages only for Dynamic Stance. The final label was assigned using a majority vote (two out of three). In cases in which the majority threshold could not be reached, the annotators held discussion sessions to reach a unanimous agreement. The average Cohen's *kappa* is 0.52. Although lower, the scores are comparable to CaSET.

Tables 1 and 2 summarize the annotated data for CaSET, while Table 3 reports the Dynamic Stance annotations for CaSERa and DuST. Concerning the Static Stance labels of the CaSET dataset, we can observe that there is a relatively balanced presence of all the labels, but the *NA* label takes almost a third of the data. By contrast, the Dynamic Stance labels of the CaSET dataset (Table 2) have a very unbalanced distribution, although we barely find messages labeled as *Unrelated* (1.52%) or *NA* (0.38%). Given that the messages are the same in both datasets, this difference comes from modeling the stance toward a static, prefixed topic, or modeling it toward other messages thus capturing a variety of attitudes. As claimed in Section 2, the empiric results show that the *Agree* label is not common in online debates.

The distribution of the Dynamic Stance labels across the three datasets is quite similar. *Agree*, *Query* and *Unrelated* are infrequent, and most interactions are labeled with *Disagree*, *Elaborate* or *Neutral*. CaSET is the dataset with more disagreements, while CaSERa is the one with more neutral interactions. This makes sense since we specifically selected controversial topics for CaSET. The different nature of the sources (i.e. micro-blogging and forums) can also have an impact on the different distribution. As for DuST, the debate around vaccines seems to spark slightly fewer disagreements and more reinforcements of the same opinions than in Catalan, with the most frequent class

| Label | Vaccines | Rent regulation | Airport expansion | Surrogate pregnancy | TV show rigging | Total |
|---|---|---|---|---|---|---|
| **Favour** | 982 (22.78%) | 990 (36.26%) | 91 (4.83%) | 160 (10.84%) | 75 (10.9%) | 2,298 (20.74%) |
| **Against** | 532 (12.34%) | 405 (14.84%) | 474 (25.31%) | 712 (48.24%) | 27 (3.92%) | 2,150 (19.41%) |
| **Neutral** | 1,806 (41.89%) | 433 (15.86%) | 487 (26.0%) | 293 (19.85%) | 242 (35.17%) | 3,261 (29.44%) |
| **NA** | 991 (22.99%) | 902 (33.04%) | 821 (43.83%) | 311 (21.07%) | 344 (50.0%) | 3,369 (30.41%) |
| **Total** | **4,311** | **2,730** | **1,873** | **1,476** | **688** | **11,078** |

Table 1: Statistics of the Static Stance annotations in CaSET. All the numbers refer to individual messages, which can correspond to parent messages or to replies. The percentages refer to the frequency of the label in the topic.

| Label | Vaccines | Rent regulation | Airport expansion | Surrogate pregnancy | TV show rigging | Total |
|---|---|---|---|---|---|---|
| **Agree** | 52 (1.81%) | 35 (2.25%) | 20 (1.86%) | 9 (1.04%) | 3 (0.75%) | 119 (1.76%) |
| **Disagree** | 1,154 (40.21%) | 620 (39.85%) | 447 (41.47%) | 466 (53.69%) | 125 (31.17%) | 2,812 (41.52%) |
| **Elaborate** | 771 (26.86%) | 621 (39.91%) | 419 (38.87%) | 288 (33.18%) | 183 (45.64 %) | 2,282 (33.70%) |
| **Query** | 157 (5.47%) | 26 (1.67%) | 28 (2.6%) | 21 (2.42%) | 8 (2.0%) | 240 (3.54%) |
| **Neutral** | 699 (24.36%) | 192 (12.34%) | 148 (13.73%) | 78 (8.99%) | 74 (18.45%) | 1,191 (17.58%) |
| **Unrelated** | 17 (0.59%) | 61 (3.92%) | 11 (1.02%) | 6 (0.69%) | 8 (2.0%) | 103 (1.52%) |
| **NA** | 20 (0.7%) | 1 (0.06%) | 5 (0.46%) | 0 | 0 | 26 (0.38%) |
| **Total** | **2,870** | **1,556** | **1,078** | **868** | **401** | **6,773** |

Table 2: Statistics of the Dynamic Stance annotations in CaSET. All the numbers are based on message pairs. The percentages refer to the frequency of the label in the topic.

being *Elaborate*.

| Label | CaSERa | DuST |
|---|---|---|
| **Agree** | 218 (1.55%) | 29 (1.17%) |
| **Disagree** | 3,183 (22.74%) | 822 (33.08%) |
| **Elaborate** | 3,992 (28.52%) | 884 (35.57%) |
| **Query** | 859 (6.14%) | 170 (6.84%) |
| **Neutral** | 5,564 (39.75%) | 574 (23.1%) |
| **Unrelated** | 175 (1.25%) | 5 (0.2%) |
| **NA** | 9 (0.06%) | 1 (0.04%) |
| **Total** | 14,000 | 2,485 |

Table 3: Label distribution for CaSERa and DuST.

| Parent → Reply → | Fav. Agai. | Agai. Fav. | Fav. Fav. | Agai. Agai. | Neut. Neut. |
|---|---|---|---|---|---|
| **Agree** | 0 | 0 | 32 | 19 | 27 |
| **Disagree** | 313 | 169 | 110 | 59 | 234 |
| **Elaborate** | 27 | 33 | 244 | 204 | 221 |
| **Query** | 8 | 3 | 10 | 3 | 49 |
| **Neutral** | 12 | 11 | 30 | 7 | 215 |
| **Unrelated** | 0 | 0 | 0 | 0 | 0 |
| **NA** | 0 | 0 | 0 | 0 | 0 |
| **TOTAL** | 360 | 216 | 426 | 292 | 746 |

Table 4: CaSET distribution of Static Stance labels per parent and reply message pair against their Dynamic Stance annotations.

Restricting the analysis to the topic distribution in CaSET, the highest rate of disagreements is in the discussion around the legalization of surrogate pregnancy. This topic is also the one with the highest number of messages against it. Rent regulation is the topic with more messages in favor. For the COVID-19 vaccination, the difference between messages in favor and against is smaller than in the other politically controversial topics, with *Neutral* being the largest class. Finally, half of the messages about the TV show rigging are not on the rigging itself, but mostly on other topics surrounding the involved artists. The observations raised by simply comparing the two annotation levels of this data are not very intuitive. For this reason, we analyzed these relations further.

**Static *vs*. Dynamic Stance** A closer look at the relation between the two levels of annotation can be seen in Table 4, in which we report the Dynamic Stance values by pairs of Static Stance values for the parent and the reply messages. While the pairs with different Static Stance (either *Favour-Against* or *Against-Favour*) are mostly labelled as *Disagree*, many pairs of messages with the same Static Stance labels (e.g. *Favour-Favour*) are also labelled as *Disagree*.

Looking at these cases we observe that they are messages that in fact disagree with each other, but without being in disagreement with the central topic (in the following examples, vaccination). Examples (1) and (2) illustrate two of these instances.

The examples have been extracted from CaSET and translated into English.

(1)  *Favour$_{Parent}$-Favour$_{Reply}$-Disagree$_{Dynamic}$*

    a.  **Parent:** How many people will die and infect new people every day that passes with vaccination stopped? 10,000(vaccine_emoji)=1(heart_emoji).

    b.  **Reply**: Covid is the disease to fight. A vaccine is the shield to eradicating it, and it is given to a healthy person. If any type of incident is recorded and is as serious as death, it must be investigated.

(2)  *Neutral$_{Parent}$-Neutral$_{Reply}$-Disagree$_{Dynamic}$*

    a.  **Parent:** On the one hand, no one can force them to get vaccinated. On the other hand, no one can be forced to allow people who refuse to be vaccinated into their business. Which rights must be respected? Serious question. In the USA, unvaccinated children are not allowed to go to public school.

    b.  **Reply**: That's a lie. They don't vaccinate children here nor in the USA.

The fact that these disagreements would be missed by the Static Stance annotation scheme, and the fact that these examples are very common (31.37% of the *Neutral$_{Parent}$-Neutral$_{Reply}$* pairs and 25.88% of the *Favour$_{Parent}$-Favour$_{Reply}$* ones), shows that the Static Stance is not enough to model the interactions of a debate. For tasks such as controversy detection, this phenomenon might arise from disagreements in specific issues of the topic, which might not be reflected by the *Against*/*Favour* scheme. In addition, for monitoring the discourse around a topic, being able to classify the interactions would provide deeper knowledge of the discussion.

## 5 Automatic Stance Detection

We developed several baseline models for both Static and Dynamic Stance detection. We investigate the abilities of pre-trained language models to model Static and Dynamic Stance with two settings: one where the topic is seen at training time and another where the topic is excluded. Lastly, we investigate the cross-lingual transfer abilities of a multilingual model on DuST.

**Experimental setup** For our experiments, we fine-tuned both the Catalan language model RoBERTa-large-ca-v2[9] (Armengol-Estapé et al., 2021) and the multilingual model mDeBERTa-v3-base[10] (He et al., 2021) for sequence classification. We used a learning rate of 1-e5, a batch size of 8, a warmup of 0.06, and trained for 10 epochs with an Adam optimizer and cross-entropy loss. We kept the best checkpoint on the development set. To deal with the issue that some labels in the Dynamic Stance annotation have very few instances, we merged the label *Agree* with *Elaborate*, *Query* with *Neutral*, and *Unrelated* with *NA*. That way, we had 4 labels for both Dynamic and Static Stance.

**All topics experiments** For our monolingual experiments, we generated fixed train, development, and test sets from CaSET. For the Static Stance, we have 9,073 instances in train and maintained 1,000 instances each for development and test. For the Dynamic Stance, we used 4,771 instances for training and 1,000 instances each for development and test. The CaSERa data has been used as additional training materials, increasing the training set to 18,771.

Table 5 shows the resulting macro F1 score for the models. In both cases, the monolingual model achieves the best scores with RoBERTa-large. For the Dynamic Stance, we observe a positive impact of the additional training materials from CaSERa, even if it comes from a different platform and with no specific topic. Although not directly comparable, the higher scores for Static Stance are in line with the IAA scores, which show that Static Stance is an easier task to model.

| Model | Static Stance Detection (CaSET) | Dynamic Stance Detection | |
| --- | --- | --- | --- |
| | | (CaSET) | (CaSET+CaSERa) |
| RoBERTa-l-ca | **0.71** | 0.62 | **0.65** |
| mDeBERTaV3 | 0.66 | 0.43 | 0.48 |

Table 5: Results of RoBERTa-l-ca and mDeBERTaV3 models for Static and Dynamic Stance in Catalan with all topics on the training set. Scores correspond to macro F1. The best scores per type of stance are in bold.

In Table 6 we report the results of the best system per label for both Static and Dynamic Stance. While labels that clearly express a stance (*Favour*, *Against*, and *Neutral*) obtain comparable results for both phenomena, the largest difference affects the *NA* label. This is likely due to differences in the training instances, with the Dynamic Stance having very few *NA* cases.

[9]https://huggingface.co/projecte-aina/roberta-base-ca-v2

[10]https://huggingface.co/microsoft/mdeberta-v3-base

| Stance Type | Label | Label F1 | Macro F1 |
|---|---|---|---|
| Static | Favour | 0.66 | **0.71** |
| | Against | 0.65 | |
| | Neutral | 0.55 | |
| | NA | 0.88 | |
| Dynamic | Disagree | 0.68 | **0.65** |
| | Elaborate | 0.67 | |
| | Neutral | 0.58 | |
| | NA | 0.56 | |

Table 6: Results per label of the models with best performance (in bold in Table 5) for Static and Dynamic Stance in Catalan.

**Zero-shot topic experiments**   One of the motivations for creating a Dynamic Stance annotation scheme is the possibility to improve the cross-topic portability of models. To test this, we conducted a series of zero-shot topic experiments which exclude one topic at a time from the training set and use it as the test set. Therefore, the sizes of the test and training sets change in each of the experiments, the details about these sets can be found in Appendix B. We have experimented only with RoBERTa-large, since it is the model that obtained the best results when using all the topics. The results are illustrated in Table 7.

We observe that, while the cross-topic performance of Dynamic Stance drops a bit from the model that had seen all topics, it still learns. This is not the case with the Static Stance cross-topic models, which have a below-chance performance in most topics. Although not directly comparable, these results suggest that Dynamic Stance can be modeled across topics in an easier way than Static Stance. It has to be noted that the addition of the CaSERa data to augment the dynamic training set does not have much effect on the results.

| Test Topic | Static Stance Detection (CaSET) | Dynamic Stance Detection (CaSET) | (CaSET+CaSERa) |
|---|---|---|---|
| Vaccines | 0.10 | 0.43 | 0.47 |
| Rent regulation | 0.14 | 0.40 | 0.44 |
| Airport expansion | 0.18 | 0.49 | 0.54 |
| Surrogate pregnancy | 0.41 | 0.47 | 0.48 |
| TV show rigging | 0.24 | 0.43 | 0.43 |

Table 7: Results of the zero-shot topic experiments in Catalan. "Test Topic" indicates the topic used at test time and excluded from the training set. Scores correspond to macro F1.

**Cross-lingual experiments**   For the cross-lingual experiments, we employed only data from CaSET

covering the vaccine topic, thus maximising the compatibility with DuST. In all of our experiments, we used mDeBERTaV3. To investigate the benefit of cross-lingual data for Dynamic Stance classification, we first developed two baselines using the Catalan and the Dutch data only. For Catalan, we used a training set of 1,868 instances and a development and test sets of 500 instances each. For Dutch, we have 1,483 instances in the training, and 500 each for development and test. For the cross-lingual experiments, we concatenated the training materials.

Table 8 reports the results. In both languages, the use of the language-specific training data obtains limited results, especially for the Dutch data. However, when concatenating the two languages, results improve, with Dutch being the one benefiting the most.

| Train – Test | Macro F1 |
|---|---|
| Catalan – Catalan | 0.43 |
| Catalan+Dutch – Catalan | **0.47** |
| Dutch – Dutch | 0.19 |
| Catalan+Dutch – Dutch | **0.37** |

Table 8: Results of the cross-lingual experiments for Dynamic Stance detection. The best scores per test language are in bold.

## 6   Conclusion and Future Work

In this work, we have investigated the task of stance detection with a focus on dynamic interactions between parent messages and their replies. Our approach addressed the limitations of traditional Static Stance detection models, which heavily rely on specific topics and exhibit poor generalization capabilities. By introducing a novel annotation scheme and creating the Dynamic Stance Corpus (DySC), we provided a new perspective on the categorization of textual interactions, which captures different insights from the data. Through our data analysis, we demonstrated empirical differences between the annotation of stance in static and dynamic contexts. Furthermore, DySC is fully developed for non-English languages, namely Catalan and Dutch.

Using a generic monolingual pre-trained model for Catalan (RoBERTa-ca-l) we achieve a macro F1 score of 0.65 for Dynamic Stance using all topics on training. We also show that, while Dynamic Stance models exhibit some learning on the zero-

shot topic scenario (with an average score of 0.47), Static Stance models go from a macro F1 score of 0.71 with all topics on train, to an average score of 0.21 when in the zero-shot scenario. We therefore conclude that Dynamic Stance is easier to model in cross-topic scenarios. As for the cross-lingual experiments, the results show that there is some knowledge transfer between languages. While the results leave room for improvement, they are close to the ones obtained by similar previous work such as the Stancesaurus (reported in Section 2).

This study represents an important initial step toward effectively modeling the task of stance detection in dynamic textual interactions. Future work will address some of the pending issues by expanding the annotated data to other languages and investigating new methods (including the use of in-context learning) to improve the portability of the Dynamic Stance models.

## Limitations

We have used a manually-curated selection of keywords for our data collection. While we have ensured to cover relevant keywords (including synonyms) for all of the topics we have targeted, the list of keywords may not be exhaustive and contain potential bias. We aim to expand the data collection by using few-shot in-context learning approaches to extend the list of keywords. Furthermore, we have to acknowledge some intrinsic limitations of the Twitter API, which prevents the collection of all potentially relevant messages.

The data collection and annotation have been conducted with all possible human resources available and applying fair treatment to all parties involved. We leave the collection of additional messages and the expansion to other languages for future work. Our guidelines are made available and so is the code we have used to collect the messages,[11] thus offering opportunities to other interested researchers to expand DySC and set up a larger benchmark for Dynamic Stance.

In our experiments, we did not use all of the fine-grained classes we have identified for the annotation. This does not mean that these labels should be discarded. Our decision was motivated by the relatively few instances we were able to identify.

Our models leave room for improvement. Our experiments represent an implementation of base-

lines to validate the annotations and identify lower-bound thresholds. The use of general language pre-trained models, RoBERTa-ca-l and mDeBERTaV3, may have had an impact on the results, considering the different language varieties of DySC. Future work will need to address this issue.

## Acknowledgements

We acknowledge Racó Català for the release of their data for this project. This work has been partially funded by Generalitat de Catalunya, within the framework of Projecte AINA. Blanca Calvo is funded by Disargue (TED2021-130810B-C21) funded by MCIN/AEI /10.13039/501100011033 and by European Union NextGenerationEU/ PRTR, and by a PIF UPV/EHU 2023 doctoral grant.

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

# A Keywords and time frames for data collection

| Topic | Keywords | Time frames |
|---|---|---|
| COVID-19 vaccines | vacunes, vacuna, vaccí, vaccines, vaccinat, vacunades, pfizer, astrazeneca, vacuna moderna, passaport covid, tercera dosis, efectes, secundaris, dosi reforç, vacunats, tercera dosis, vacuneu-vos, vacunem-nos, vacunet, vacune't | 1st January - 31st December 2021 |
| Rent regulation | regulació rent, preus de rent, regularrents, regulació dels preus, okupes, RegulemElsrentsJA, desnonaments, desnonament, fons voltor, fons voltors, seguretat als propietaris, propietaris immobiliaris, immobiliari, blackstone, lobbies immobiliaris, llei de vivenda | 1st January - 31st December 2020 |
| Airport expansion | ampliació airport, #noampliacioairport, 19S, airport delta, airport el prat, sostenible airport, més verd del món, delta llobregat, soroll avions, airport internacional, ricarda, ambiental airport | 1st August - 31st October 2021 |
| Surrogate pregnancy | gestació surrogate, ventre de rent, mares portadores, inscripció de menors, dona gestant, gestaciosurrogate, paternitat surrogate, ventres de rent, comprar nens, embaràs subrogat, inscripció de nens, mare gestant, comprar criatures, pares biològics, comprar bebes, filiar un nadó, explotació reproductiva, maternitat surrogate | 30th January - 6th February 2017 1st August - 16th November 2017 1st February - 24th April 2019 30th August - 6th September 2020 26th February - 27th March 2022 |
| TV show rigging | benidorm fest, tanxungueiras, chanel, rigoberta, ay mama hai fronteiras, slomo, rtve jurat, eurovisión, tongo | 29th January - 15st February 2022 |

Table A.1: Keywords and time frames used to retrieve Twitter messages in CaSET.

| Topic | Keywords | Time frame |
|---|---|---|
| COVID-19 vaccines | vaccins, vaccin, vaccinatie, mRNA, pfizer, astrazeneca, sputnik, moderna, 2G, 3G, vaccinatiebewijs, coronatoeganbewijs, gevaccineerd, vaccineren, prikken, prik, inenten, vaccineer, eerste prik, tweede prik, booster, booster vaccinatie, derde prik, boostervaccinatie, coronabewijs, coronavaccins, coronavaccinatie | 1st September - 31st December 2021 |

Table A.2: Keywords and time frame used to retrieve Twitter messages in DuST.

# B Zero-shot topic experiments

(a) Dynamic Stance

| Test topic | Train (CaSET only) | Train (CaSET+CaSERa) | Dev | Test |
|---|---|---|---|---|
| vaccines | 3,403 | 17,403 | 500 | 2,870 |
| rent | 4,717 | 18,717 | 500 | 1,556 |
| airport | 5,195 | 19,195 | 500 | 1,078 |
| surrogate | 5,405 | 19,405 | 500 | 868 |
| benidormfest | 5,872 | 19,872 | 500 | 401 |

(b) Static Stance

| Test topic | Train (CaSET only) | Dev | Test |
|---|---|---|---|
| vaccines | 6,267 | 500 | 4,308 |
| rent | 7,844 | 500 | 2,731 |
| airport | 8,678 | 500 | 1,897 |
| surrogate | 9,138 | 500 | 1,437 |
| benidormfest | 9,873 | 500 | 702 |

Table B.1: Size of the sets in the Zero-shot topic experiments.