# OpenReview forum: "Dynamic Stance: Modeling Discussions by Labeling the Interactions"
_EMNLP/2023/Conference — EMNLP 2023 Findings_

### Official Review · Reviewer_m8Kb · 2023-07-27

**Soundness:** 4

**Excitement:**

4: Strong: This paper deepens the understanding of some phenomenon or lowers the barriers to an existing research direction.

**Paper Topic And Main Contributions:**

The paper presents a new way of modeling the task of stance detection accompanied by an annotated dataset in Dutch and Catalan encompassing several different topics and initial experiments testing the potential for topic and language transfer. In contrast to traditional stance detection, the new task models stance as a dynamic relationship between two texts (in the case of the current dataset between a target and a parent post in an online discussion). The underlying motivation behind this approach is that dynamic stance detection is more generalizable across topics than static stance detection, which always relies on a pre-defined topic (or claim) with respect to which texts are assessed. Initial experiments with language models confirm that the new model of the task does indeed lead to better topic transfer.

I found the paper well-written, well-argued, and easy to follow. I think it makes a valuable contribution to the field in presenting a new way of modeling stance detection together with a new resource and experiments and should be accepted to the conference. I do have some doubts about the comparison between static and dynamic stance annotation (see comments below),  but I think they can most likely be clarified in the final version of the paper and should not prevent acceptance.

**Reasons To Accept:**

The paper presents a new way of modeling the tasks of stance detection dynamically instead of statically. I see this as a promising new way of framing the task.

Together with the new task, the paper presents a new resource in Catalan and Dutch annotated for static and dynamic stance detection. This is a rich new dataset that allows for analyzing static and dynamic stance annotations.

The paper presents initial experiments with language models that give valuable first insights into the generalizability of the two annotation frameworks.

**Reasons To Reject:**

I did not see real reasons to reject the paper. There are, however, a couple of points that should be clarified:

Annotation analysis: I am really not so sure about the comparison of static versus dynamic stance annotations and their consequences for the data (Section 4.2). The annotations resulting from both annotation frameworks are considered side by side and it is discussed what aspects of a debate they can capture. My problem with this comparison is that the two annotation frameworks essentially capture a different task; one (static) assesses a text with respect to a claim or topic (predefined, explicit) while the other (dynamic) assesses a text with respect to another text. In the latter, the claim in the first text remains entirely implicit. While the overall topic of a discussion may be related to the topic of covid vaccines, the first text may contain a claim that is entirely unrelated. This is not really an insight that arises from the data. Rather, it is inherent to the design of the two different frameworks. I think it is valid to show the consequences of these differences in the data, but I think it is important to be clear about the fact that the frameworks have different goals and thus capture different information.

Update after rebuttal: Thank you for the clarification. I am confident that the authors can make the necessary changes in the final version of the paper.

Automatic stance detection: Please clarify whether the train and test splits for dynamic and static stance detection are the same. I understand that there is a difference in the number of instances (and labels) due to the different annotation frameworks, but right now, it is not clear whether the texts are the same. It would be great to have a bit more information about topic overlap between train and text in the ‘all topics experiments’.

Update after rebuttal: Thank you for the clarification!

Topic independence is a huge plus of the dynamic way of modeling stance. I think it would be good to make this (and its consequence for automatic approaches) more prominent in the framing of the paper.

Minor remark: I don’t think that Dutch can really be called a low-resource language with respect to NLP resources. ‘Low resource’ may be true with respect to stance resources, but not in terms of general NLP resources (see for instance Joshi et al. 2020). Perhaps check this.

Reference
Joshi, P., Santy, S., Budhiraja, A., Bali, K. and Choudhury, M., 2020, July. The State and Fate of Linguistic Diversity and Inclusion in the NLP World. In Proceedings of the 58th Annual Meeting of the Association for Computational Linguistics (pp. 6282-6293).

**Reproducibility:**

4: Could mostly reproduce the results, but there may be some variation because of sample variance or minor variations in their interpretation of the protocol or method.

**Reviewer Confidence:**

3: Pretty sure, but there's a chance I missed something. Although I have a good feel for this area in general, I did not carefully check the paper's details, e.g., the math, experimental design, or novelty.

---

> ### Author Rebuttal · Authors · 2023-08-25
>
> Dear R3, thank you for the review and comments. Concerning your comments:
> - Annotation analysis: We do agree that the two annotation frameworks are different tasks that capture different insights. This is mentioned in line 180, but we will make it clearer since R2 also had doubts regarding this. As stated in line 181, we annotate both levels to investigate the interactions in a descriptive way, which in turn exemplifies that Dynamic Stance can be used in many discussions, even if they are not on the specified topic. We will make it clear that the frameworks have different goals and thus capture different information.
> - Automatic stance detection: Regarding the test subsets used for Dynamic and Static Stance, I can confirm that they are the same subsets. However, given that the Dynamic approach is modeled by pairs of messages, there are fewer instances in both the training and the test set.  Regarding the “All topic experiments”, we have added a table in the final version that addresses the distribution of topics in these subsets.
> - Thanks for pointing out the contribution by Joshi et al. (2020). Our wording was not the best. Given that Joshi et al. (2020) consider Dutch as a good example of the "Underdog" class (lots of unsupervised data, few labeled data), and given the results of the EU project European Langauge Equality 2 of both Catalan and Dutch, we will reword the presentation of both languages as middle-resourced languages.

---

### Official Review · Reviewer_Q591 · 2023-07-28

**Soundness:** 3

**Excitement:**

3: Ambivalent: It has merits (e.g., it reports state-of-the-art results, the idea is nice), but there are key weaknesses (e.g., it describes incremental work), and it can significantly benefit from another round of revision. However, I won't object to accepting it if my co-reviewers champion it.

**Paper Topic And Main Contributions:**

This paper is about classifying whether a reply agrees or disagrees with the preceding text. The authors propose to differentiate this task (“dynamic” stance detection)  from the task of “static” stance detection, which is independent from any preceding text. (It is unclear whether the authors conceptualize dynamic stance detection as a stand-alone task, or as an objective that should be learnt jointly with static stance detection.)
The authors annotate social media posts on several controversial topics in Catalan and Dutch. They then present some automatic classification results for the proposed task, based on fine-tuning a pre-trained encoder model.

**Reasons To Accept:**

- Useful resource: The paper presents an annotated dataset in two languages and on several topics, while ensuring cross-lingual comparability.
- Baseline results: The paper presents basic experiments, including on cross-topic and cross-lingual transfer, that will be useful as a reference for future work.

**Reasons To Reject:**

Unfortunately, the presentation of the research questions and the results is not very clear, and especially the relation to “static” stance detection remains unclear: Does this paper propose an improvement over previous approaches, or does it propose an entirely new task?
- Unclear presentation: The experiments with automatic stance detection (Section 5) focus on a comparison between “Static Training” and “Dynamic Training”. However, the two concepts are not formally defined. Is “Dynamic Training” multi-task learning on dynamic and static stance detection? If so, the use of different subsets used in the experiments (L496–501) make it difficult to see whether this objective has a positive effect. Or is “Dynamic Training” just learning to classify the relation to the parent text, independent of the static stance? In that case, it is not clear in what respect this approach “address[es] the limitations of traditional Static Stance detection” (L569), since it would appear to be a new task.
- Claim not supported by example: A main claim of the paper (L420f.) is that static stance and dynamic stance are not equal: Two statements can have the same stance w.r.t a topic but still disagree with each other. While Table 4 provides quantitative support for this claim, Example 1 (L437–446) does not seem to be a valid illustration. In fact, Reply (b) seems to be critical of the COVID-19 vaccination campaign, and not in favor. I would encourage the authors to check the example and make sure it does not contain an annotation error.

**Reproducibility:**

5: Could easily reproduce the results.

**Reviewer Confidence:**

4: Quite sure. I tried to check the important points carefully. It's unlikely, though conceivable, that I missed something that should affect my ratings.

**Typos Grammar Style And Presentation Improvements:**

- Thanks for the paper, it was interesting to read. Personally, I found it challenging to develop an intuition for “dynamic stance”, because “dynamic” usually means something that is changing over time. Sections 3 and 4 provide more clarity, but maybe intuition could also be helped by an alternative word choice.
- L206: “intelligible” -> “unintelligible”
- L392: “Disagre” -> “Disagree”

---

> ### Author Rebuttal · Authors · 2023-08-25
>
> Dear R2, thank you for the review and comments. Concerning your comments:
> - Research question: As stated in line 180 and also highlighted by R3, we present Dynamic Stance detection, a new stand-alone task with an accompanying annotated dataset. In Section 2, we motivate our task and identify the differences with the state of the art (lines 12-16; lines 137-146; lines 161-166). Additionally, as stated in line 182, we annotate Static Stance to draw the relation between these two frameworks, as the latter has been the most used approach for stance detection in the last few years. Our main claim from the comparison is that, while Static Stance has the issue of being difficult to generalize to new topics, Dynamic Stance can do this easier. Therefore, this new and independent way of modeling stance can be used for new topics for which we do not have any annotated data. We will make this aspect clearer in the Introduction and Conclusion. We will also acknowledge that these two independent tasks capture slightly different insights from the input.
> - Static Training / Dynamic Training: We agree that the wording is not optimal, although it only affects Table 5 and Table 7. In Section 5, we present the experiments as Static Stance Detection and Dynamic Stance Detection. In the final version, we will change the titles in Table 5 and Table 7 to “Static Stance Detection” and “Dynamic Stance Detection”. In both cases, this is a text classification task and the models were trained by fine-tuning Transformer models. In the Static Stance detection model, the input was the target text. In the Dynamic Stance detection model, the input was the parent message paired with the reply.
> - Test set split: The test subsets used for Static and Dynamic Stance Detection are the same. However, given that the Dynamic approach is modeled by pairs of messages, there are fewer instances in both the training and the test set.
> - Claim not supported by example: As we have answered to R1, we disagree. We have checked and this is not an annotation error, but rather an interpretation error. Here is our full-fledged explanation: the Reply does not express a disagreement against the vaccination policy in general (“A vaccine is the barrier to eradicating it”), rather it is a call of caution and for an investigation of suspicious deaths (which is exactly what the EU did with the AstraZeneca vaccines). However, this call to be cautious (which is still in favor of vaccines as they are “the barrier to eradicating the disease”) disagrees with the original message which requests a fast pace with the vaccination program. When seen in isolation, all our annotators labeled this message as being in Favour of vaccines. However, when looking at the dynamic stance, two annotators labeled the interaction as Disagree, and two of them labeled it as Neutral. The judge, however, rightly assessed that the messages were disagreeing. It is a non-trivial example, but it perfectly illustrates our task, the difficulty of the annotation, and the difference between static stance and its dynamic counterpart. Given that two reviewers misread this example, we will extend the explanation of this case.
> - The use of the term “Dynamic” is motivated to differentiate the standard stance detection tasks (what we call “Static”), in which texts are labeled towards a fixed topic or claim, from the new proposed task, in which every text is labeled towards a previous text and can be about any topic. Therefore, the reference claim becomes dynamic i.e. it can be a different one in every instance.
> - Finally, thank you for pointing out the two typos in the paper, they have already been changed for the final version.

---

### Official Review · Reviewer_yDAp · 2023-08-02

**Soundness:** 3

**Excitement:**

3: Ambivalent: It has merits (e.g., it reports state-of-the-art results, the idea is nice), but there are key weaknesses (e.g., it describes incremental work), and it can significantly benefit from another round of revision. However, I won't object to accepting it if my co-reviewers champion it.

**Paper Topic And Main Contributions:**

The paper describes two new datasets for dynamic stance annotation in two languages, Dutch and Catalan. It's shown that a cross-lingual approach to stance detection helps performance for Dutch, for which there is only a smaller dataset available.

**Questions For The Authors:**

l09: describe what you mean by parent messages or use another term

l233ff: wrt the Query label -- this label seems to be solely driven by the grammatical structure of the sentence, instead of its function in the discourse. Reading the first sentence of the description `expressing doubts about the content' goes somewhat against that because it's more general.

l256ff: The example oversimplifies the stance here: 'More poison for us' is not a disagreement with the previous message (rather with the implied content that more Covid vaccines is a good thing). It's rather `expressing doubts about the content'. The characterisation of 'Disagree' lists dislike of the parent message in the case of rhetorical questioning, which is not the case in the example here. --> tighten up the descriptions of the labels

Section 4.2. - given stance annotation might be quite subjective, it'd be good to know in how many cases adjudication had to be performed to arrive at the gold labels.

Example 1b: I don't see how in the static stance, this is Favour_Reply... Maybe adjust the English translation in 1a, it's not really clear what it means...

l490: merging the labels 'Query' with 'Neutral' seems counterintuitive based on the descriptions above...

l545ff: What do you think is the reason for the improvement?



**Reasons To Accept:**

There is merit in the fact that the authors annotate both static and dynamic stance in order to show the relations between the two levels of analysis. I also like the crosslinguistic approach in Catalan and Dutch.

**Reasons To Reject:**

The paper has a couple of gaps that I address below in the more detailed comments. Overall I think this is rather an LREC paper, given its focus.

**Reproducibility:**

3: Could reproduce the results with some difficulty. The settings of parameters are underspecified or subjectively determined; the training/evaluation data are not widely available.

**Reviewer Confidence:**

4: Quite sure. I tried to check the important points carefully. It's unlikely, though conceivable, that I missed something that should affect my ratings.

---

> ### Author Rebuttal · Authors · 2023-08-25
>
> Dear R1, thank you for the review and comments. Concerning your comments:
> - LREC paper: We disagree with this comment. We present a new task (see also R3) in the Sentiment Analysis track and a new resource. The paper fits EMNLP’s criteria expressed in the CfP.
> - “parent message”: We agree that this wording can be confusing before the framework is explained in Section 3. We will reword as follows in the Abstract: “interaction between a message and their replies”. Furthermore, in Section 3, we will add to L207 the following clarification: “To capture the dynamic dimension of stance, the target message must always be put in relation to its direct parent, that means, the message that the target message is replying to”.
> - Query definition: The label “Query” in the Dynamic Stance framework is not grammatical, but rather discursive. In our annotation guidelines, the label is defined as follows: “A reply will be annotated as Query if the message questions or requires more arguments/proof/evidence about the parent message. It won’t be annotated as Query if the question is rhetorical, ironic, or has a clear stance towards the topic of the parent message.”  The definition of the label Disagree in the annotation guidelines also specifies that “If the reply questions the parent message and it shows a clear disagreement with it (e.g. use of irony, rhetorical questions, etc.), the reply will be labeled as Disagree.” Additionally, the guidelines contain examples that show the difference between Query and Disagree. We will include these clearer and more specific definitions in the paper and the full guidelines will be published with the final version.
> - L248: Following your observations, we have realized that the sentence preceding the “More poison for us” example is not specific enough. We will change it to “In our guidelines, we emphasize that, in the cases in which the parent message does not hold an opinion, the label of the reply must be assigned on the basis of the content." Therefore, “More poison for us” exemplifies a reply to a parent message with no stance associated (“EMA approves another COVID vaccine”), in which, as emphasized in the paragraph, the label must be assigned precisely based on the content of the parent message. In this case, the reply disagrees with the content of the parent message (“these vaccines are poison, and this new vaccine is another poison”), i.e., it shows a clear dislike of the content of the parent message.
> - Section 4.2 role of the judge: Thanks for this comment. We will add the following sentence after line 343: “That means that the judges were involved in 1,284 cases for the Dynamic Stance annotation (18.95% of the instances), and 2,087 for the Static Stance annotation (18.84% of the instances).” And, in line 356, we will add: “The judge was involved in 18.17% of the cases.”
> - Example 1b: We have checked and this is not an annotation error, but rather an interpretation error (R2 also had issues with this example). Here is our full-fledged explanation: the Reply does not express a disagreement against the vaccination policy in general (“A vaccine is the barrier to eradicating it”), rather it is a call of caution and for an investigation of suspicious deaths (which is exactly what the EU did with the AstraZeneca vaccines). However, this call to be cautious (which is still in favor of vaccines as they are “the barrier to eradicating the disease”) disagrees with the original message which requests a fast pace with the vaccination program. When seen in isolation, all our annotators labeled this message as being in Favour of vaccines. However, when looking at the dynamic stance, two annotators labeled the interaction as Disagree, and two of them labeled it as Neutral. The judge, however, rightly assessed that the messages were disagreeing. It is a non-trivial example, but it perfectly illustrates our task, the difficulty of the annotation, and the difference between static stance and its dynamic counterpart. Given that two reviewers misread this example, we will extend the explanation of this case.
> - Merge of “Query” and “Neutral”: When it comes to the stance of the reply towards the parent message, “Query” and “Neutral” labels do not express any clear stance, thus merging them is not counterintuitive.
> - Improvement in cross-lingual experiments: For us, this is clearly due to a larger amount of training instances, together with the fact they are on the same topic and use the same annotation scheme.

---

### Meta-Review · Area_Chair_2LiY · 2023-09-26

**Recommendation:** 3

**Metareview:**

The paper describes two new datasets for dynamic stance annotation in two languages, Dutch and Catalan. It's shown that a cross-lingual approach to stance detection helps performance for Dutch, for which there is only a smaller dataset available.

The reviewers were generally in agreement that the use of annotations of both static and dynamic stance to show relations on different levels of analysis was a strong aspect of the paper, and a framework that could be useful in future work. The cross linguistic approach was also very well received. Further the paper offers a useful resource to the community and provides strong baseline results.

The reasons to reject that concern me most (which are not resolved in the rebuttal/discussion period) include:
* the suggestion that the paper may be more suited for LREC rather than a conference focusing on empirical analyses
* the unclear presentation of the thesis of the paper (which was discussed in the rebuttal, but I do see the point the reviewer makes)

I feel this paper is strong and could be a useful resource to the community. I do also feel that the authors should address the issues brought up by reviewers and adapt the paper accordingly before acceptance.

---

### Decision · Program_Chairs · 2023-10-07

**Decision:**

Accept-Findings

**Comment:**

The paper describes two new datasets for dynamic stance annotation in two languages, Dutch and Catalan. It's shown that a cross-lingual approach to stance detection helps performance for Dutch, for which there is only a smaller dataset available.

The reviewers were generally in agreement that the use of annotations of both static and dynamic stance to show relations on different levels of analysis was a strong aspect of the paper, and a framework that could be useful in future work. The cross linguistic approach was also very well received. Further the paper offers a useful resource to the community and provides strong baseline results.

The reasons to reject that concern me most (which are not resolved in the rebuttal/discussion period) include:
* the suggestion that the paper may be more suited for LREC rather than a conference focusing on empirical analyses
* the unclear presentation of the thesis of the paper (which was discussed in the rebuttal, but I do see the point the reviewer makes)

I feel this paper is strong and could be a useful resource to the community. I do also feel that the authors should address the issues brought up by reviewers and adapt the paper accordingly before acceptance.